# Soluble SIGLEC5: A New Prognosis Marker in Colorectal Cancer Patients

**DOI:** 10.3390/cancers13153896

**Published:** 2021-08-02

**Authors:** Karla Montalbán-Hernández, Ramón Cantero-Cid, Roberto Lozano-Rodríguez, Alejandro Pascual-Iglesias, José Avendaño-Ortiz, José Carlos Casalvilla-Dueñas, Gloria Cristina Bonel Pérez, Jenny Guevara, Cristóbal Marcano, Cristina Barragán, Jaime Valentín, Carlos del Fresno, Luis Augusto Aguirre, Eduardo López Collazo

**Affiliations:** 1The Innate Immune Response Group, IdiPAZ, La Paz University Hospital, 28046 Madrid, Spain; karlamarina.montalban@salud.madrid.org (K.M.-H.); ramon.cantero@salud.madrid.org (R.C.-C.); roberto.lozano.rodriguez@idipaz.es (R.L.-R.); alejandro.pascual.iglesias@idipaz.es (A.P.-I.); jose.avendano@idipaz.es (J.A.-O.); josecarlos.casalvilla.duenas@idipaz.es (J.C.C.-D.); gloria.bonel.perez@idipaz.es (G.C.B.P.); jennyrosario.guevara@salud.madrid.org (J.G.); cristobalsimon.marcano@salud.madrid.org (C.M.); cristina.barragan@hgvillalba.es (C.B.); jaime.valentin.quiroga@idipaz.es (J.V.); carlos.delfresno.sanchez@idipaz.es (C.d.F.); luis.augusto.aguirre@idipaz.es (L.A.A.); 2Tumor Immunology Lab, IdiPAZ, La Paz University Hospital, 28046 Madrid, Spain; 3Centre for Biomedical Research Network of Respiratory Diseases (CIBERES), 28029 Madrid, Spain

**Keywords:** colorectal cancer, sSIGLEC5, survival, prognosis, predictive

## Abstract

**Simple Summary:**

Amongst colorectal cancers, there is significant heterogeneity, which hinders the search for a single disease detection approach. Clinical prognostic markers are urgently needed. The aim of our prospective study was to analyse the possible role of pre-operative soluble SIGLEC5 plasma levels in patient prognosis and evolution. In a cohort of 114 patients with colorectal cancer, our data confirmed the relevance of soluble SIGLEC5 levels as a prognosis marker and exitus predictor. Altogether, our data indicate that levels of this protein could be a novel and promising biomarker for patients with colorectal cancer.

**Abstract:**

Colorectal cancer (CRC) is the second most deadly and third most commonly diagnosed cancer worldwide. There is significant heterogeneity among patients with CRC, which hinders the search for a standard approach for the detection of this disease. Therefore, the identification of robust prognostic markers for patients with CRC represents an urgent clinical need. In search of such biomarkers, a total of 114 patients with colorectal cancer and 67 healthy participants were studied. Soluble SIGLEC5 (sSIGLEC5) levels were higher in plasma from patients with CRC compared with healthy volunteers. Additionally, sSIGLEC5 levels were higher in exitus than in survivors, and the receiver operating characteristic curve analysis revealed sSIGLEC5 to be an exitus predictor (area under the curve 0.853; cut-off > 412.6 ng/mL) in these patients. A Kaplan–Meier analysis showed that patients with high levels of sSIGLEC5 had significantly shorter overall survival (hazard ratio 15.68; 95% CI 4.571–53.81; *p* ≤ 0.0001) than those with lower sSIGLEC5 levels. Our study suggests that sSIGLEC5 is a soluble prognosis marker and exitus predictor in CRC.

## 1. Introduction

Colorectal cancer (CRC) incidence has steadily increased in the developed world, whilst the age of disease onset has decreased over the years [1]. The heterogeneity observed amongst the various CRCs makes it difficult to generate a single, user-friendly approach towards disease detection. In the last decade, a number of potential markers have been studied, including carcinoembryonic antigen, carbohydrate antigen, tissue polypeptide specific antigen and tumor-associated glycoprotein-72. However, the diagnostic accuracy of all of these parameters is limited [2].

Although most research is focused on revealing intrinsic factors that control cancer progression, a number of teams have shifted towards the study of the tumor microenvironment, and particularly the role of the immune system [3]. Along these lines, the number of circulating CD8^+^ T lymphocytes has been strongly correlated with a better prognosis in patients with CRC [4]. In addition, the number of tumor-infiltrating T lymphocytes (TILs) has been shown to be inversely correlated with disease stage [5]. These data highlight how immune surveillance is favored in the early disease stages and becomes compromised in advanced cancers. Laghi et al., 2009 also described how a high density of CD3^+^ TILs in node-negative cancers, in which no tumor infiltration to the lymph nodes occurred, led to a significantly lower chance of generating metachronous metastasis [6]. However, this predictive value was insufficient in node-positive cancers, illustrating the importance of tumor immune evasion [6]. Antigenic presentation is crucial for T cell activation, but this process can become inhibited in the tumor microenvironment by the expression of immune checkpoints (ICs) on the surface of the cancer cells, leading to immune evasion and T cell anergy [7].

CRC has one of the highest mutation prevalence rates amongst all cancers [8], producing a high neo-antigen load. One reason why immunotherapy is not as effective in CRC as in other cancers might be the existence of potent immunosuppressive capabilities in CRC [9]. These highly immunogenic tumors have led to the design of several ongoing IC inhibitor trials in this disease [10]. Taube et al., 2014 state that the immune evasion is not driven by the well-established programmed cell death protein 1 (PD-1)/programmed cell death ligand 1 (PD-L1) pathway and that the combination of chemotherapy with immunotherapy could lead to more effective treatment regimens [11,12]. However, trial results have demonstrated that this effectiveness might apply only to certain CRC tumors, showing promising results with pembrolizumab (αPD-1) in CRCs with microsatellite instability (MSI) [13]. Although CRCs with MSI only comprise a small percentage of all CRCs, these findings shed light on the advances in cancer immunotherapy and demonstrate the importance and potential usefulness of IC inhibitors in CRC treatment.

Among immune-regulatory molecules, sialic acid-binding immunoglobulin-type lectins (SIGLECs) are mostly expressed by immune cells (TILs, natural killers, dendritic cells, and macrophages), and they regulate functions of both the innate and adaptive immune system, which makes them highly interesting as ICs [14,15,16]. These lectins display an amino-terminal V-set immunoglobulin domain that recognizes sialic acids [17]. It is noteworthy that this domain mediates the interaction of some ICs and their ligands, such as PD-1 and PD-L1 [18]. SIGLECs have been studied for over 40 years as possible immunotherapy targets in hematopoietic cancers [14,19]. Hypersialylation of cancer cells has been recognized as a hallmark of poor prognosis due to the recruitment of inhibitory SIGLECs that suppress the immune response via interaction with sialic acids [20,21,22]. Among this receptor family, SIGLEC5 and its soluble form, sSIGLEC5, have been reported to block the highly sialylated p-selectin glycoprotein ligand-1 (PSGL-1) molecule on leucocytes, enhancing anti-inflammatory activity and supporting a pro-tumoral environment [23,24]. Moreover, SIGLEC5 has recently been found increased on TILs of various cancers, suggesting its contribution towards T cell exhaustion [25].

Identification of predictive markers of CRC progression has become critical in clinical practice. Soluble markers, such as sCD137, sE-cadherin, and sCXCL16 have already been studied and are interesting markers in CRC; however, none have enough supportive data to pursue these as routine clinical markers or therapeutic targets [26,27,28]. To address this gap, we hypothesized that sSIGLEC5 might have predictive value for CRC progression. Herein, we found sSIGLEC5 concentrations to be higher in plasma of patients with CRC compared with healthy volunteers, suggesting a potential role for sSIGLEC5 as a disease detection marker. Furthermore, we found that sSIGLEC5 levels positively correlated with CRC stage. Ultimately, sSIGLEC5 has been shown to be a good prognostic marker and exitus predictor for CRC in a prospective study. These findings open new paths towards the use of sSIGLEC5 as a biomarker in CRC progression.

## 2. Materials and Methods

### 2.1. Patient and Healthy Volunteer Recruitment

Patients diagnosed with CRC (*n* = 114) were enrolled in this study prior to surgery at La Paz University Hospital (Madrid, Spain). CRC was diagnosed by colonoscopy and tumor biopsy. Patients were followed-up a median time of 1.75 years and were classified according to their disease stage (Table 1). Three patients were excluded from the survival analysis due to noncancer-related death during their follow-up. Blood samples were taken 24 h before patients entered the operating room; peripheral blood mononuclear cells and plasma were isolated following standardised procedures [29]. As controls, healthy volunteers (*n* = 67) were recruited from the Blood Donor Services of La Paz University Hospital (Table 1).

### 2.2. ELISA Assay

Concentrations of sSIGLEC5 in plasma samples from patients with colorectal cancer and healthy volunteers were determined using a commercially available enzyme-linked immunosorbent assay (ELISA) kit (Sigma-Aldrich, St Louis, MI, USA), following the manufacturer’s instructions.

### 2.3. Soluble Immune Checkpoint Measurement

Soluble ICs (sPD-1and sPD-L1) were measured in plasma samples from patients with colorectal cancer (*n* = 114) and healthy volunteers (*n* = 30) using the LegendPlex Custom Human Immune Checkpoint Panel, following the manufacturer’s instructions (Biolegend, San Diego, CA, USA). Briefly, plasma samples were incubated with premixed capture antibody-coated beads, washed, incubated with detection antibodies and streptavidin with phycoerythrin conjugate, acquired using a FACSCalibur flow cytometer (BD) and analyzed with Biolegend v8.0 software (Biolegend, San Diego, CA, USA).

### 2.4. Statistical Analysis

Continuous variables were analysed by either parametric or nonparametric statistical tests after analyzing their Gaussian distribution by the Shapiro–Wilk test. Categorical parameters were calculated as proportions and were analysed by chi-squared testing. A one-way analysis of variance with Tukey’s correction test was used to analyse the comparison between disease stages. Pearson’s r correlations and linear regression were used to evaluate the correlation between variables. Contingency table analysis was performed with chi-squared test. For the analysis of overall survival, we used Kaplan–Meier survival curves and receiver operating characteristic (ROC) analyses (GraphPad Prism 9). The Cox Mantel log-rank test model and Gehan–Breslow–Wilcoxon test were applied for outcome prediction, and the Wilson/Brown test was used for ROC analyses. Positive and negative predictive values were calculated with the Yardstick R Package (version 403). Regression model analyses were conducted with SPSS version 23 (IBM) software. In addition, Microsoft Office Excel and GraphPad Prism 9 were used to calculate the Youden index; the area under the curve (AUC) and 95% confidence intervals (CIs) are reported. A value of *p* < 0.05 was considered significant.

### 2.5. Ethics Approval

All patients and controls signed informed consent documents, and the data were treated according to recommended criteria of confidentiality, following the ethical guidelines of the 1975 Declaration of Helsinki. The study was approved by the local Ethics Committee (La Paz University Hospital, Madrid, PI-1958, approval date: 8 April 2015).

## 3. Results

### 3.1. Patient Characteristics

From 28 May 2015 to 11 August 2020, a total of 114 patients with CRC were consecutively recruited by the Digestive Surgery Service of La Paz University Hospital in Madrid, Spain. Blood was collected 24 h prior to surgery. Patients were followed-up until 7 April 2021 with continuous check-ups, and were classified into four groups according to their disease stage: I, II, III, or IV. Table 1 summarizes the patients’ characteristics, including: tumor and metastasis site, comorbidities, American Society of Anesthesiologists score and patient outcome. As a control group, healthy volunteers (HVs, *n* = 67) were recruited by the Blood Donor Services of La Paz University Hospital and were also assessed in this cohort.

### 3.2. sSIGLEC5 Is a Prognosis Biomarker in Colorectal Cancer Patients

The concentrations of sSIGLEC5 were determined in pre-operative plasma samples from all participants. Figure 1 shows significantly higher sSIGLEC5 levels in patients with CRC (*n* = 114) compared with HVs (*n* = 67) (*p* ≤ 0.0001). Note exitus patients (in red) show the highest values of sSIGLEC5. On the other hand, other soluble ICs (sPD-1 and sPD-L1) commonly assayed for CRC immunotherapies did not show significant differences in plasma concentration between the patients with CRC (*n* = 114) and randomly chosen HVs (*n* = 30), as shown in Appendix A.

To study the role of pre-operative sSIGLEC5 level as a potential prognosis biomarker, patients were classified according to their disease stage (Table 1). Figure 2A shows that levels of sSIGLEC5 trend upwards in the three higher CRC stages compared with CRC stage I. Note that exitus patients (in red) showed sSIGLEC5 levels higher than the average for their disease stage. In addition, the plasma concentrations of sSIGLEC5 showed a positive correlation with the patients’ disease stage (Figure 2B; Pearson r = 0.2595, *p* = 0.005). Similar results were obtained when correlations between pre-operative plasma sSIGLEC5 levels and both tumor dedifferentiation grade and lymph node infiltration were analyzed (Appendix A).

### 3.3. sSIGLEC5 Is an Exitus Predictor in Patients with Colorectal Cancer

Given that pre-operative levels of sSIGLEC5 in exitus were found to be higher than the average for the patients with advanced disease stage (Figure 2A), we explored the potential value of sSIGLEC5 plasma levels as an exitus predictor in CRC. To differentiate between survivor and exitus, a ROC analysis of the plasma sSIGLEC5 levels was performed (Figure 3A, AUC = 0.853; 95% CI 0.7729–0.9344; *p* = 0.0001). The optimal cut-off value, estimated by the Youden index, was 412.6 ng/mL and exhibited a high sensitivity (1; 95% CI 74.12–100), and a specificity of 0.67 (95% CI 57.31–75.44). The contingency chart showed a significantly higher probability of exitus in the sSIGLEC5-high group compared with the sSIGLEC5-low group (Figure 3B, χ^2^ = 17.91, *p* < 0.0001). The calculated optimal cut-off distinguished survivors from exitus (*p* ≤ 0.0001), as Figure 3C shows. Appendix A also shows exitus (in red) for sPD-1 and sPD-L1, with no significant discrimination regarding survivors for these markers.

Note, the use of TNM or therapeutic information did not generate a patent predictive model to discriminate survivors from exitus (Appendix A). In this regard, both of them could lead to generate either false positives or miss true positives. In contrast, pre-operative sSIGLEC5 levels showed a robust predictor value which allowed strong discrimination of exitus patients.

To study whether sSIGLEC5 served as an independent survival factor in CRC, a multivariate binary regression was done. The Wald forward conditional regression model was carried out to establish whether the variables: age, sex, disease stage, treatment, tumor size, metastasis, node infiltration, lymphovascular invasion, perineural invasion, surgical margins, Carcinoembryogenic Antigen (CEA), tumor dedifferentiation and sSIGLEC5 levels had an independent effect on exitus. The final model after two steps (Table 2) showed only dedifferentiation and sSIGLEC5 levels as unique variables which exhibited an independent association to mortality. Appendix A illustrates those variables which were excluded in step 2 of the model and their *p* values. Moreover, both tumor cell dedifferentiation and sSIGLEC5 levels also exhibited a significant positive correlation (Appendix A).

Wald forward conditional stepwise regression, including age, gender, disease stage, treatment, tumor size, metastasis, node infiltration, lymphovascular invasion, perineural invasion, surgical margins, CEA, tumor dedifferentiation, and sSIGLEC5 levels as variables. Final model after two steps is shown. Units: tumor dedifferentiation in grading system and sSIGLEC5 levels in ng/mL. B, B weight coefficient; OR, odds ratio; OR CI 95, 95% confidence interval of odds ratio; SD, standard deviation of B; Wald, Wald statistic.

### 3.4. High sSIGLEC5 Concentration Is Associated with Decreased Overall Survival in Patients with Colorectal Cancer

Patients with colorectal cancer were classified into sSIGLEC5-High and sSIGLEC5-Low groups according to their Youden cut-off value prior to surgery. A Kaplan–Meier analysis showed patients belonging to the sSIGLEC5-High group had a significantly shorter overall survival than those in the sSIGLEC5-Low group, indicating the potential of sSIGLEC5 as an exitus predictor in patients with CRC (Figure 4). Both statistical analyses performed, the Gehan–Breslow–Wilcoxon and the log-rank (Mantel-cox), showed statistical significance. Survival time was normalized to a median follow-up because patients entered the study at various time points. Survival proportions for the median follow-up of the study, 1.75 years, showed a shorter overall survival in the sSIGLEC5-High group (77%) compared with the sSIGLEC5-Low group (100%). The predictive strength of sSIGLEC5 was evaluated based on the concordance with our validated Youden index. We evaluated the accuracy, sensitivity, specificity, positive predictive value (PPV) and negative predictive value (NPV) for sSIGLEC5. The likelihood ratio (LR) shows how much more likely a patient or HV is to receive a positive result, based on Youden’s index, in exitus compared with a survivor. The PPV for sSIGLEC5 in exitus was 24% and the NPV was 100% (LR^+^ 2.86–LR^−^ 0). These values confirmed the robustness of the predictive model.

## 4. Discussion

Although patients with CRC have a long-life expectancy, the five-year survival rate decreases from 90% in stages I/II to 12% in advanced stages III/IV, mainly due to metastasis development in the later stages [30]. Notably, strong biomarkers are missing for colorectal cancer prognosis due to the heterogeneity of this disease, and chemotherapy still remains the gold standard option for advanced CRC stages. In this regard, the need for novel disease biomarkers has become a priority issue.

It was not until 2020 that the United States Food and Drug Administration (FDA) approved the first IC treatment in CRC [31], pembrolizumab (αPD-1), for the treatment of CRC with high MSI. Recent clinical trials have shown that, despite the highest clinical efficacy observed in melanoma and lung cancer, αPD-1 or αPD-L1 therapies are less effective in CRC; moreover, they are restricted to CRC with high MSI [32]. Nonetheless, the latter only correspond to a limited percentage of all CRC forms, emphasizing the need for new IC targets. According to our data, soluble levels of these two ICs (PD-1 and PD-L1) did not show significant differences between the HVs and the patients with CRC studied, nor did they serve as exitus predictors.

Herein, we have reported an incremented amount of pre-operative sSIGLEC5 in plasma samples of patients with CRC, showing concentration values significantly higher in the patients with CRC with a poorer outcome. In addition, plasma levels of SIGLEC5 correlated with disease stage, tumor dedifferentiation and lymph node infiltration, all which point to its efficacy as a biomarker of CRC evolution.

Moreover, a ROC analysis revealed a robust AUC for sSIGLEC5 as an exitus predictor in CRC. The cut-off value calculated based on the Youden index disclosed that levels above 412.6 ng/mL of sSIGLEC5 indicate a poor outcome. Our findings show sSIGLEC5 levels will help distinguish those patients at risk of a poorer progression or exitus. The positive and negative predictive values obtained also highlight the robustness of our cut-off value. These results could provide a feasible value to incorporate into routine patient blood sample analyses before surgery. In addition, the logistic regression model supports the role for sSIGLEC5 levels and tumor dedifferentiation grade as independent survival factors in CRC. Note, tumor dedifferentiation is established through an invasive procedure: biopsy, and the grading has a range of bias. In contrast, the quantification of sSIGLEC5 levels is not subjective to interpretation and the procedure is minimally invasive. Altogether, our data postulates pre-operative sSIGLEC5 levels as a novel alternative prognosis marker and strong exitus predictor in CRC, with a pre-operative value strongly associated with overall survival.

It is worth considering that pre-operative sSIGLEC5 level might not only be a biomarker or exitus predictor, but also a potential IC candidate in CRC, with a specific role in disease evolution. As explained earlier, the SIGLEC5 structure exhibits strong similarities with that from well-known IC candidates, and it has recently been described as a patent IC candidate in sepsis [33]. The hypersialylation present in the cancer cells might be recruiting and interacting with sSIGLEC5, helping the tumor cells escape from immune surveillance. In this regard, further experiments will be necessary to establish both the origin of sSIGLEC5 and to define its mechanism of action in CRC. Meanwhile, our data suggest the usefulness of this soluble marker as a prognostic and exitus predictor in CRC.

## 5. Conclusions

In conclusion, our data show pre-operative sSIGLEC5 levels are elevated in patients with CRC and how levels of this protein increase with disease stage, demonstrating its power as a novel disease prognosis biomarker. Additionally, levels of sSIGLEC5 were significantly higher in exitus patients compared with survivors, showing its patent role as an exitus predictor in CRC. A cut-off value of 412.6 ng/mL, with a high sensitivity (1) and specificity (0.67), allowed for this strong discrimination between survivors and exitus. Altogether, our data demonstrate the importance that pre-operative plasma sSIGLEC5 levels could have in predicting the patient’s outcome and, with additional experimentation, sSIGLEC5 could become a target for therapy for CRC.

## Figures and Tables

**Figure 1 cancers-13-03896-f001:**
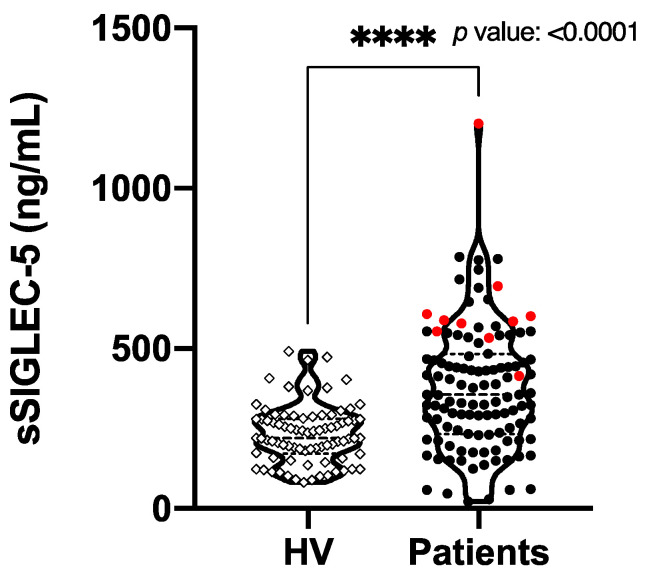
Pre-operative soluble SIGLEC5 is elevated in patients with colorectal cancer vs. healthy volunteers. The soluble SIGLEC5 (sSIGLEC5) levels in plasma from healthy volunteers (*n* = 67) and patients with colorectal cancer (*n* = 114) quantified by enzyme-linked immunosorbent assay are shown. The difference between groups was analyzed by an unpaired Mann–Whitney U test, 95% confidence interval. A *p*-value < 0.05 was used as the level of significance, **** *p* < 0.0001.

**Figure 2 cancers-13-03896-f002:**
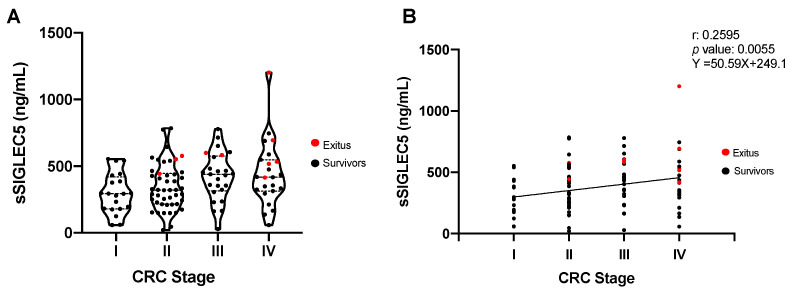
Pre-operative soluble SIGLEC5 levels correlated with disease stage in patients with colorectal cancer. (**A**) Levels of sSIGLEC5 in plasma from patients with colorectal cancer classified into stage I–IV are shown (*n* = 114). The differences between stages were analyzed by a one-way analysis of variance with Tukey’s correction test. (**B**) Correlation between plasma sSIGLEC5 levels and disease stage in patients with colorectal cancer is shown. The correlation between sSIGLEC5 and disease stages was analysed by Pearson’s r. Simple correlations of every stage were performed with 95% confidence intervals. A *p*-value < 0.05 was used as the level of significance.

**Figure 3 cancers-13-03896-f003:**
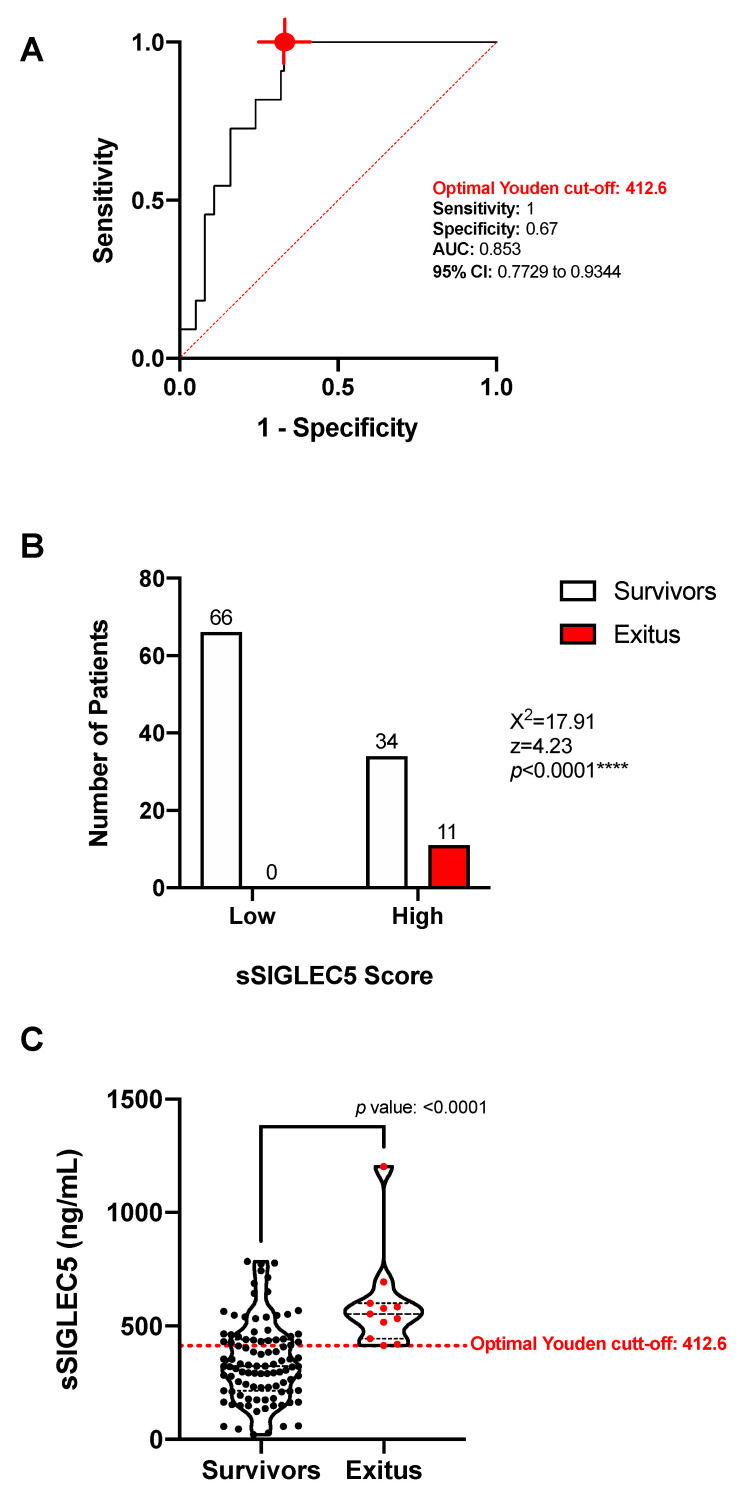
Pre-operative soluble SIGLEC5 (sSIGLEC5) levels predict exitus in patients with colorectal cancer. (**A**) The receiver-operating-characteristic (ROC) curve describing the predictive performance value of plasma sSIGLEC5 for exitus in patients with colorectal cancer (*n* = 111) (red line; area under the curve (AUC), 0.853 (95% CI 0.7729*–*0.9344)) is shown. Optimal cut-off, estimated by the Youden index for plasmatic sSIGLEC5 concentration, 412.6 ng/mL (red point shows optimal Youden cut-off specificity and sensitivity values). The ROC curve analysis was performed by Wilson/Brown test, with 95% confidence interval. (**B**) Chart shows how exitus is significantly higher in the sSIGLEC5-high group compared with the sSIGLEC-low group. The contingency table was analyzed with the chi-squared test. (**C**) The sSIGLEC5 levels in plasma in the survivor (*n* = 100) and exitus (*n* = 11) groups separated by the optimal Youden index (red dashed line) are shown. Differences in sSIGLEC5 levels between survivor or exitus groups were analyzed by Mann–Whitney U test. A *p*-value < 0.05 was used as the level of significance. **** *p* < 0.0001.

**Figure 4 cancers-13-03896-f004:**
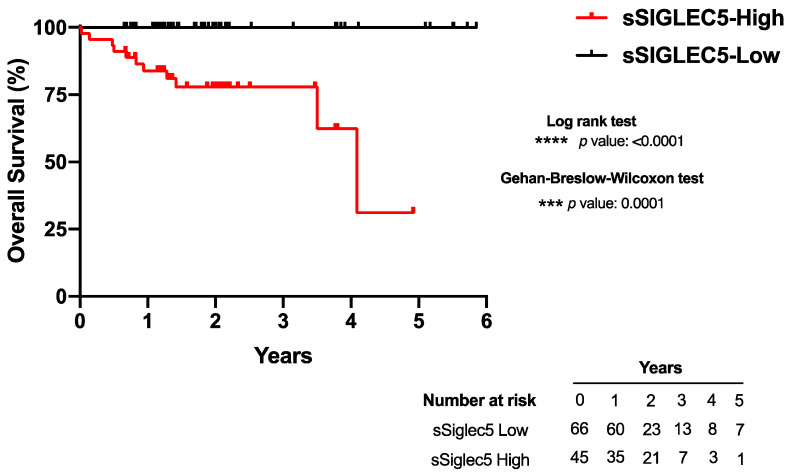
Pre-operative soluble SIGLEC5 levels predict survival of patients with colorectal cancer. Kaplan–Meier survival curve from surgery date to study end date, 7 April 2021, according to baseline plasma sSIGLEC5 is shown. The differences between survival rates associated with sSIGLEC5 were calculated by a log-rank (Mantel-Cox; *p*
*<* 0.0001) test with 95% confidence interval. The Gehan–Breslow–Wilcoxon test also showed similar significance levels at early time points (*p*
*=* 0.0001). A *p*-value < 0.05 was used as the level of significance. *** *p* < 0.001; **** *p* < 0.0001.

**Table 1 cancers-13-03896-t001:** Patient and HV characteristics ^1^.

Characteristic	Healthy Volunteers *n* = 67	All Patients *n* = 114	Stage I *n* = 20	Stage II *n* = 47	Stage III *n* = 25	Stage IV *n* = 22	*p*-Value
Sex							0.804
Male	29 (43)	71 (56)	11 (55)	26 (55)	16 (64)	11 (50)	
Female	38 (56)	54 (43)	9 (45)	21 (44)	9 (36)	11 (50)	
Age							0.268
Median Range	59 (50–75)	71 (50–92)	70,5 (50–86)	75,5 (52–91)	70 (59–92)	69,5 (51–88)	
Metastasis		35 (30)	0 (0)	2 (5)	5 (20)	22 (100)	<0.0001 (****)
Synchronic		22 (62)	0 (0)	0 (0)	0 (0)	22 (100)	<0.0001 (****)
Metachronic		13 (37)	0 (0)	2 (100)	5 (100)	6 (26)	0.006 (**)
Site of Metastasis							
Liver		25 (71)	0 (0)	1 (4)	3 (12)	21 (84)	<0.0001 (****)
Lung		9 (25)	0 (0)	1 (12)	2 (25)	6 (66)	0.001 (**)
Peritoneum		2 (5)	0 (0)	0 (0)	0 (0)	2 (100)	0.036 (*)
Exitus		11 (10)	0 (0)	3 (27)	2 (18)	6 (54)	0.013 (*)
Tumor Site							
Caecum		18 (15)	6 (33)	9 (50)	2 (11)	1 (5)	0.083
Ascending Colon		30 (26)	6 (20)	13 (43)	7 (23)	4 (13)	0.808
Transverse Colon		9 (7)	2 (22)	5 (55)	1 (11)	1 (11)	0.689
Descending Colon		45 (39)	4 (8)	17 (37)	10 (22)	14 (31)	0.032 (*)
Rectum		11 (9)	2 (18)	3 (27)	5 (45)	1 (9)	0.230
Comorbidities							
Smoker		51 (44)	5 (9)	24 (47)	11 (21)	11 (21)	0.244
Arterial Hypertension		64 (56)	12 (18)	28 (43)	14 (21)	10 (15)	0.713
Dyslipidaemia		42 (36)	7 (16)	22 (52)	9 (21)	4 (9)	0.148
Diabetes Mellitus		27 (23)	5 (18)	8 (29)	10 (37)	4 (14)	0.156
BMI							
Median		25.9	27.1	25.5	25.95	23.45	
Range		(18.4–43.7)	(18.4–43.7)	(22.3–43.7)	(18.4–37)	(19.4–32)	0.041 (*)
ASA Score							
I		4 (3)	1 (25)	2 (50)	0 (0)	1 (25)	0.756
II		43 (37)	9 (20)	14 (32)	9 (20)	11 (25)	0.368
III		63 (55)	10 (15)	29 (46)	16 (25)	8 (12)	0.174
IV		4 (4)	0 (0)	2 (50)	0 (0)	2 (50)	0.291

^1^ Data are presented as number (%) or median (range). *p* values show significant differences between patient groups. * *p* < 0.05; ** *p* < 0.01; **** *p* < 0.0001.

**Table 2 cancers-13-03896-t002:** Logistic regression model for exitus prediction in colorectal cancer.

Variable	B	SD	Wald	*p*-Value	OR	OR CI 95	
						Low	High
Dedifferentiation	3.977	1.335	8.880	0.00288	53.343	3.901	729.508
sSIGLEC5	0.010	0.004	6.303	0.01205	1.009	1.002	1.017

## Data Availability

No new data were created or analyzed in this study. Data sharing is not applicable to this article.

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
