# Peer review of "Soluble SIGLEC5: A New Prognosis Marker in Colorectal Cancer Patients"

_cancers, 2021, doi:10.3390/cancers13153896_

Round 1
Reviewer 1 Report
The authors showed that sSIGLEC5 levels correlate with the CRC stage and predict survival. Despite the lack of formal evidence of the source of sSIGLEC5 and mechanistic role in CRC, the authors claim that this molecule could be used as a biomarker routine test to anticipate prognosis in CRC.
While the study is purely descriptive it would be highly valuable to address the consequences of sSIGLEC5 on leukocyte populations, by profiling the blood of naïve patients in the presence of sSIGLEC5-enriched plasma, to address the direct role of this soluble molecule on T lymphocytes or other myeloid cells (neutrophils and monocytes).
Minor comments:
1. Could the authors comment on the modulation of sSIGLEC5 levels in the plasma of responders’ vs non-responders to ICB in CRC?
2. Does sSIGLEC5 levels act as a prognostic marker and exitus predictor in other solid tumors?
Author Response
We welcome your questions and recommendations. Please find a point by point response in the attached document.

Reviewer 2 Report
Manuscript entitled "Soluble SIGLEC5: A new prognosis marker in colorectal cancer patients"
This work contains few information and is less clinically relevant. It should be improved by:
- Correlate Soluble SIGLEC5 to detail clinicicopathologic features such as tumor differentiation, vascular/perineurial invasion, surgical margin status, CEA level, ... etc.
- For the survival analysis, the above mentioned features as well as pT, pN, therapeutic information (C/T, R/T, and more) should also be added into univariate and multivariate analysis to see whether Soluble SIGLEC5 confers an independent survival factor.
- The Soluble SIGLEC5 is overlapped between HV and CRC patients, indicating it is not an markers for detection. If this marker can not add any independent survival impact, it is of very low clinical value.
Author Response

(The authors gave the same response as above.)

Round 2
Reviewer 2 Report
Revision of manuscript entitled "Soluble SIGLEC5: A new prognosis marker in colorectal cancer patients"
- The survival analysis for important parameter should be conducted for more detail. For example, the therapeutic information should be described and analyzed in more detail.
- The statistic work should be improved. for example, the nodal meta., and other categorical parameter should not be analyzed in this way.
- Overall, the quality of this work didn't improved too much.
Author Response
Dear Reviewer#2, please find a point-by-point answer to your suggestions in the attached document.

Round 3
Reviewer 2 Report
I think it it ready for publication now.